# A large-scale genome-wide association and meta-analysis identified four novel susceptibility loci for leprosy

Zhenzhen Wang[1,2,*], Yonghu Sun[1,2,*], Xi'an Fu[1,2,3,*], Gongqi Yu[1,2,4,*], Chuan Wang[1,2], Fangfang Bao[1,2], Zhenhua Yue[1,2,3], Jianke Li[1,2,5], Lele Sun[1,2], Astrid Irwanto[6], Yongxiang Yu[1,2], Mingfei Chen[1,2], Zihao Mi[1,2], Honglei Wang[1,2,3], Pengcheng Huai[1,2,3], Yi Li[6], Tiantian Du[1,2,5], Wenjun Yu[1,2,5], Yang Xia[1,2,5], Hailu Xiao[1,2], Jiabao You[1,2], Jinghui Li[1,2], Qing Yang[1,5], Na Wang[1,2,3], Panpan Shang[1,2], Guiye Niu[1,2], Xiaojun Chi[1,2,5], Xiuhuan Wang[1,2,5], Jing Cao[1,2,3], Xiujun Cheng[1,2,3], Hong Liu[1,2,5], Jianjun Liu[6] & Furen Zhang[1,2,3,4,5,7]

Leprosy, a chronic infectious disease, results from the uncultivable pathogen *Mycobacterium leprae* (*M. leprae*), and usually progresses to peripheral neuropathy and permanent progressive deformity if not treated. Previously published genetic studies have identified 18 gene/loci significantly associated with leprosy at the genome-wide significant level. However as a complex disease, only a small proportion of leprosy risk could be explained by those gene/loci. To further identify more susceptibility gene/loci, we hereby performed a three-stage GWAS comprising 8,156 leprosy patients and 15,610 controls of Chinese ancestry. Four novel loci were identified including rs6807915 on 3p25.2 ($P = 1.94 \times 10^{-8}$, OR = 0.89), rs4720118 on 7p14.3 ($P = 3.85 \times 10^{-10}$, OR = 1.16), rs55894533 on 8p23.1 ($P = 5.07 \times 10^{-11}$, OR = 1.15) and rs10100465 on 8q24.11 ($P = 2.85 \times 10^{-11}$, OR = 0.85). Altogether, these findings have provided new insight and significantly expanded our understanding of the genetic basis of leprosy.

[1] Shandong Provincial Institute of Dermatology and Venereology, Shandong Academy of Medical Sciences, Jinan, Shandong 250000, China. [2] Shandong Provincial Key Laboratory for Dermatovenereology, Jinan, Shandong 250000, China. [3] School of Medicine, Shandong University, Jinan, Shandong 250000, China. [4] School of Medicine and Life Science, University of Jinan-Shandong Academy of Medical Sciences, Jinan, Shandong 250022, China. [5] Shandong Provincial Hospital for Skin Diseases, Shandong University, Jinan, Shandong 250000, China. [6] Human Genetics, Genome Institute of Singapore, Singapore 138672, Singapore. [7] National Clinical Key Project of Dermatology and Venereology, Jinan, Shandong 250000, China. * These authors contributed equally to this work. Correspondence and requests for materials should be addressed to H.L. (email: hongyue2519@hotmail.com) or to F.Z. (email: zhangfuren@hotmail.com).

Leprosy, an ancient mycobacterial disease, results from the uncultivable pathogen *Mycobacterium leprae*, and usually progresses to peripheral neuropathy and permanent progressive deformity if not being treated[1]. The clinical features of leprosy present a five-group spectrum, likely reflecting the interactive outcome between the host immune responses and the pathogen. Since the implementation of multidrug therapy, the prevalence of leprosy has been significantly reduced worldwide. However, due to the occurrence of permanent disabilities and sequelae, leprosy still represents a serious health problem in the developing countries[2].

The role of host genetic factors in the development of leprosy has been well established through epidemiological and molecular genetic studies. Multiple genes and loci have been discovered as leprosy risk factors, such as *PARK2-PACRG* (ref. 3), *IL10* (ref. 4), *VDR* (ref. 5), *LTA* (ref. 6) and *HLA-DR* (ref. 7), but only a few were replicated. Recently, the understanding of leprosy genetic factors has been remarkably improved by the application of genome-wide association studies (GWASs), which have discovered 18 leprosy associated susceptibility gene/loci[8–12], most of which are related to immunity and inflammatory responses, providing valuable insights and emphasizing the important role of genetic risk factors in disease development. However, these loci can only explain a small proportion of the disease risk and heritability, indicating that additional genetic risk factors remain to be discovered.

Here, we performed a new GWAS analysis including 1,197 leprosy cases and 1,426 controls by using a population-specific array (Illumina Omni Zhonghua Array). Furthermore, we conducted a three-stage GWAS Meta analysis comprising 8,156 cases and 15,610 controls of Chinese ancestry. We confirmed all the known leprosy susceptibility loci and identified four novel loci on 3p25.2 (*SYN2*), 7p14.3 (*BBS9*), 8p23.1 (*CTSB*) and 8q24.11 (*MED30*). Altogether, these findings significantly expand the understanding of the disease susceptibility factor and suggest new biological pathways related to leprosy.

## Results

**Genome-wide discovery analysis**. To discover additional leprosy susceptibility loci, we carried out a large-scale three-stage GWAS analysis of leprosy in Chinese population. The genome-wide discovery analysis (Stage 1) involved two published GWAS dataset of leprosy[8,12], consisting of leprosy patients and geographically matched controls from northern part (Chinese Han) and southern part of China (Chinese Han and ethnic minorities), details in the Methods. The third sample was a new GWAS data set (GWAS3) of 1,197 leprosy cases and 1,426 controls from northern (Chinese Han) and southern China (Chinese Han) conducted by using Illumina Omni Zhonghua chips with 900,015 single-nucleotide polymorphisms (SNPs).

We performed the genome-wide imputation in the three GWAS data sets seperately, aiming to obtain a more comprehensive genome-widely coverage of genetic variants. The untyped SNPs were imputed by using the multi-ethnic reference panel from the 1000-genome project (March 2012 release, IMPUTE v2). Principal components analysis confirmed that all the samples were Chinese ancestry. Quality control filtering was performed to the imputed datasets as described in the Methods. Finally, we tested the associations of 5,546,030 common SNPs (minor allele frequency >1%; 258,961 genotyped, 5,287,069 imputed) in a total number of 2,743 leprosy patients and 3,573 healthy controls. Both the quantile-quantile plots (QQ plots) (Supplementary Fig. 1) and genomic inflation factors ($\lambda_{GC}$) of the genome-wide test statistic (1.026 for GWAS1, 1.019 for North Han of GWAS2, 0.98 for South Han of GWAS2, 1.052 for South Minority of GWAS2, 1.038 for North Han of GWAS3 and 1.066 for South Han of GWAS3) suggested minimal inflation on the population stratification. As shown in Fig. 1, all the 18 reported leprosy susceptibility gene/loci showed significant association in the new meta-analysis. Furthermore, additional suggestive SNPs with $P$ values $< 5 \times 10^{-4}$ from association analysis in the new GWAS or combined data sets as described in the method section were observed (Supplementary Fig. 2).

**Validation analysis of novel associations**. In total, we selected 168 top independent SNPs that met our selection criteria (Methods) for a follow-up genotyping validation analysis (Stage 2) in an independent cohort comprising 1,516 leprosy patients and 1,512 healthy controls of northern Chinese Han.

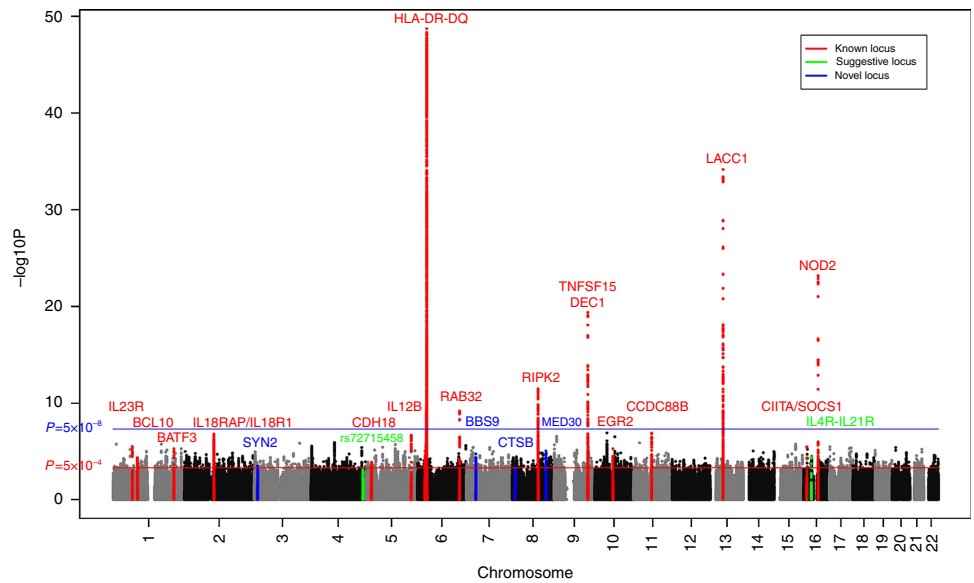

**Figure 1 | Chromosomal plot of the genome-wide association analysis.** Known loci are defined as loci previously published with genome-wide significance ($P < 5 \times 10^{-8}$); Suggestive loci are defined as loci with suggestive evidence of association ($1 \times 10^{-6} < P < 5 \times 10^{-8}$); Novel loci are defined as loci in the current study validated at genome-wide significance ($P < 5 \times 10^{-8}$).

Of the 127 successfully genotyped SNPs, we subsequently selected 21 significant SNPs with $P < 0.05$ and showing a consistent risk effect across the Stage 1 and 2 analyses for a further validation analysis (Stage 3) in four additional independent sample series from different geographic regions of China, totalling 3,897 cases and 10,525 controls (Supplementary Tables 2 and 3). The combined analysis of Stage 2 and 3 samples revealed significant association at seven SNPs after correction for multiple testing for 127 SNPs.

Totalling 8,156 leprosy patients and 15,610 healthy controls were involved in the joint association analysis (Stage 1, 2 and 3) using meta-analysis under a fixed-effects model. Four novel associations were discovered at genome-wide significance ($P < 5 \times 10^{-8}$), including rs6807915 on 3p25.2 ($P = 1.94 \times 10^{-8}$, OR = 0.89), rs4720118 on 7p14.3 ($P = 3.85 \times 10^{-10}$, OR = 1.16), rs55894533 on 8p23.1 ($P = 5.07 \times 10^{-11}$, OR = 1.15), and rs10100465 on 8q24.11 ($P = 2.85 \times 10^{-11}$, OR = 0.85) (Table 1). Two novel suggestive associations were also identified at rs72715458 on 4q34.3 ($P = 2.62 \times 10^{-7}$, OR = 0.85) and rs34411505 on 16p12.1 ($P = 5.82 \times 10^{-7}$, OR = 0.86), whose evidence was just below genome-wide significance (Table 1).

**Gene prioritization of those novel associations**. Most of the novel associations are located within polygenic LD blocks (Fig. 2). To evaluate susceptibility gene candidates within the newly confirmed loci, we performed a gene prioritization based on a differentiated gene expression analysis, which estimates the relevance gene expression between leprosy biopsy and healthy control skin through an unpublished RNA-sequence dataset (27 leprosy biopsy Vs 18 healthy controls) (Fig. 3). Those genes mostly nearby the lead variant were taken into consideration as potentially causal.

The SNP rs6807915 located between *SYN2* and *PPARG* gene, which were significantly down regulated in the lesion of leprosy patients. Although we did not find direct evidence of eQTL for rs6807915, the significant eQTL effect of four highly correlated SNPs ($r^2 > 0.9$, $D' > 0.9$, Supplementary Table 4) with it suggested that *SYN2* might more likely to be the causal gene of 3p25.2. The SNP rs4720118 located in the eighteenth intron of *BBS9* gene, which was significantly down regulated in the skin of leprosy patient. It was found that this SNP could significantly regulate the expression of *BBS9* in whole blood. The SNP rs55894533 located nearby the 5′ of *CTSB* gene, which were significantly up regulated in the leprosy patients. Significant eQTL effect of rs55894533 was found in whole blood and fibroblasts for *CTSB* gene. The SNP rs10100465 located nearby *MED30* gene, which was down-regulated in the skin of leprosy patients. Further eQTL analysis demonstrated this SNP could regulate the expression of *MED30* in whole blood.

**Fine mapping of the associations in the MHC region**. As expected, the strongest association signal in the meta-analysis was observed within the MHC region. To fine-map and elucidate the signals, we tested the association within the MHC region after imputing untyped SNPs, classical HLA alleles and polymorphic amino acid positions in the discovery data set. The significant associations within the MHC region were discovered to locate within the MHC class II region. HLA-DRB1*15 was identified as the most significant risk allele ($P = 4.21 \times 10^{-44}$; OR = 2.17). The effect of all the other associations within the MHC region could be eliminated by conditioning on HLA-DRB1*15. The full association results of classical HLA allele were provided in Supplementary Table 5.

**Heritability and enrichment analysis**. We investigated the proportion of risk variance and heritability explained by genome-

wide SNPs using Genome-wide Complex Trait Analysis (GCTA) method. We estimated the SNP heritability of leprosy at 0.199 (s.e. = 0.01), by using the genotyped autosomal SNPs and assuming the disease prevalence of leprosy as 0.0001. In total, all the identified Genome-wide significant variants thus far as being robustly associated with leprosy risk explain ∼13.53% on the liability scale (Supplementary Table 6). We conducted the heritability partitioning by tissue and functional category using LD score regression and identified significant enrichment in multiple tissue. The most significant enrichment was found in immune cells (enrichment = 3.74, $P = 3.2 \times 10^{-8}$), suggesting the important role of immunity in the disease aetiology. Region of functional category of genome with the most significant enrichment is transcription start site (TSS, enrichment = 16.3, $P = 2.52 \times 10^{-4}$) (Supplementary Fig. 3).

**Discussion**

The current large-scale GWAS meta-analysis of leprosy has several advantages comparing to our previous GWAS analysis. First, we carried out a new Genome-wide genotyping by Illumina Omni Zhonghua Chip, which is designed specific to Chinese Han population and may uncover the population-specific susceptibility loci. Second, multiple risk alleles with small effect size were detected by the increased sample size and improved statistical power. Through the meta-analysis of a total number of 8,156 leprosy patients and 15,610 healthy controls, we have identified four novel associations, all of which can indicate candidate genes within the susceptibility loci, *SYN2* on 3p25.2, *BBS9* on 7p14.3, *CTSB* on 8p23.1 and *MED30* on 8q24.11, through a differential gene expression and eQTL analysis.

At 3p25.2, we identified a non-coding variant nearby *PPARG* and *SYN2* that were both significantly down regulated in the lesion of leprosy patients. Further eQTL analysis has suggested that *SYN2* might more likely to be the causal gene than *PPARG* within the 3p25.2 locus. *SYN2* encodes neuronal phosphoproteins and belongs to the synapsin gene family, which is found to be associated with the cytoplasmic surface of synaptic vesicles. There have been several publications that support the significant association of *SYN2* with type II diabetes[13,14], interaction with BMI (ref. 15) and schizophrenia[16]. Furthermore, a down-regulated gene expression of synapsin 2 was discovered in the prefrontal cortex of schizophrenic patients[17]. Interestingly, the expression *SYN2* was also down-regulated in the leprosy biopsy in our study, which may suggest its role in the infection progress of mycobacteria to the nerves.

At 7p14.3, we identified a non-coding variant within the eighteenth intron of *BBS9* gene. It was significantly up regulated in the leprosy biopsy, while its eQTL effect was found in the thyroid and whole blood. The expression of *BBS9* could be down-regulated by parathyroid hormone in an osteoplastic cell line[18] and interrupted in a translocation breakpoint associated with Wilms Tumour[19]. Although we did not find any publication of its role on bacteria infection or autoimmunity, the fact that association and differentiated gene expression of *BBS9* gene suggest a potential role in the pathogenesis of leprosy.

At 8p23.1, Cathepsin B(CTSB) were found to be significantly up regulated in the leprosy patients. In addition, eQTL effect of the identified SNP was also observed in whole blood and fibroblasts for *CTSB* gene. *CTSB* encodes a lysosomal cysteine protease and has been reported to contribute to the progression and invasion of multiple cancer[20,21]. In the stratum spinosum of human skin, *CTSB* is found to be presented within vesicles of cellular protrusions forming cell–cell contact sites between keratinocytes[22]. It is also evident that a block in *CTSB* expression reduced the migration ability of keratinocytes and unobstructed

**Table 1 | Novel SNPs reaching genome-wide significance and suggestive SNPs approaching genome wide significance**

| Info | Minor allele/Major allele | Study | F_A | F_U | P | OR |
|---|---|---|---|---|---|---|
| rs10100465 | A/G | 1.GWAS1 | 0.27 | 0.28 | 1.32E-02 | 0.79 |
| chr8:118626279 | | 2.GWAS2 | 0.25 | 0.27 | 7.22E-02 | 0.87 |
| MED30 | | 3.GWAS3 | 0.25 | 0.29 | 1.28E-03 | 0.82 |
| | | 4.Meta-GWAS | | | 8.80E-06 | 0.82 |
| | | 5.Validation1 | 0.24 | 0.27 | 1.48E-02 | 0.87 |
| | | 6.Validation2 | | | 9.27E-06 | 0.86 |
| | | 7.All validation | | | 4.23E-07 | 0.86 |
| | | 8.Meta-analysis | I2 = 61.12; | Phet = 0.22 | 2.85E-11 | 0.85 |
| rs4720118 | T/C | 1.GWAS1 | 0.34 | 0.29 | 3.43E-03 | 1.32 |
| chr7:33469241 | | 2.GWAS2 | 0.32 | 0.29 | 3.60E-02 | 1.17 |
| BBS9 | | 3.GWAS3 | 0.34 | 0.31 | 3.79E-02 | 1.13 |
| | | 4.GWAS1+2+3 | | | 1.94E-05 | 1.20 |
| | | 5.Validation1 | 0.39 | 0.34 | 4.15E-04 | 1.20 |
| | | 6.Validation2 | | | 1.03E-03 | 1.12 |
| | | 7.All validation | | | 2.86E-06 | 1.14 |
| | | 8.Meta-analysis | I2 = 0; | Phet = 0.55 | 3.85E-10 | 1.16 |
| rs55894533 | C/A | 1.GWAS1 | 0.43 | 0.42 | 5.54E-01 | 1.06 |
| chr8:11749242 | | 2.GWAS2 | 0.46 | 0.42 | 9.10E-03 | 1.20 |
| CTSB | | 3.GWAS3 | 0.47 | 0.43 | 4.85E-04 | 1.21 |
| | | 4.GWAS1+2+3 | | | 1.14E-03 | 1.15 |
| | | 5.Validation1 | 0.44 | 0.39 | 1.14E-03 | 1.19 |
| | | 6.Validation2 | | | 2.26E-06 | 1.15 |
| | | 7.All validation | | | 1.13E-08 | 1.16 |
| | | 8.Meta-analysis | I2 = 29.82; | Phet = 0.21 | 5.07E-11 | 1.15 |
| rs6807915 | C/T | 1.GWAS1 | 0.48 | 0.48 | 3.86E-01 | 1.08 |
| chr3:12313846 | | 2.GWAS2 | 0.43 | 0.48 | 5.49E-03 | 0.82 |
| SYN2 | | 3.GWAS3 | 0.45 | 0.50 | 2.54E-04 | 0.82 |
| | | 4.GWAS1+2+3 | | | 6.71E-04 | 0.87 |
| | | 5.Validation1 | 0.47 | 0.50 | 6.28E-03 | 0.87 |
| | | 6.Validation2 | | | 2.84E-04 | 0.90 |
| | | 7.All validation | | | 6.68E-06 | 0.89 |
| | | 8.Meta-analysis | I2 = 0; | Phet = 0.95 | 1.94E-08 | 0.89 |
| rs72715458 | A/G | 1.GWAS1 | 0.14 | 0.14 | 2.43E-01 | 0.87 |
| chr4:181224835 | | 2.GWAS2 | 0.14 | 0.15 | 3.43E-01 | 0.91 |
| No gene | | 3.GWAS3 | 0.12 | 0.16 | 1.23E-05 | 0.71 |
| | | 4.GWAS1+2+3 | | | 8.01E-04 | 0.82 |
| | | 5.Validation1 | 0.12 | 0.14 | 2.34E-02 | 0.82 |
| | | 6.Validation2 | | | 9.08E-04 | 0.87 |
| | | 7.All validation | | | 7.26E-05 | 0.86 |
| | | 8.Meta-analysis | I2 = 22.38; | Phet = 0.27 | 2.62E-07 | 0.85 |
| rs34411505 | G/A | 1.GWAS1 | 0.14 | 0.14 | 2.71E-01 | 0.87 |
| chr16:27406689 | | 2.GWAS2 | 0.12 | 0.15 | 4.22E-03 | 0.74 |
| IL4R-IL21R | | 3.GWAS3 | 0.13 | 0.16 | 3.60E-02 | 0.80 |
| | | 4.GWAS1+2+3 | | | 3.97E-05 | 0.79 |
| | | 5.Validation1 | 0.14 | 0.16 | 3.74E-02 | 0.86 |
| | | 6.Validation2 | | | 8.05E-03 | 0.90 |
| | | 7.All validation | | | 8.82E-04 | 0.89 |
| | | 8.Meta-analysis | I2 = 48.6; | Phet = 0.08 | 5.82E-07 | 0.86 |

migration of keratinocytes, which could possibly explain the over expression of *CTSB* in the leprosy lesion. This remains to be further investigated.

At 8q24.11, *MED30* was identified as the potential causal gene through robust association and differentiated gene expression. *MED30* was previously reported as a suggested association with Kawasaki disease in Han Chinese population[23].

Besides these four newly identified leprosy susceptibility loci, we noted two additional suggestive associations at 4q34.3 and 16p12.1 whose evidences were both just below the genome-wide significance. The first of the two suggestive disease association rs72715458 on 4q34.3 lies within an intergenic region without any identifiable genes in the LD block. The second suggestive association rs34411505 on 16p12.1 locates between *IL4R* and *IL21R* whose expressions were both elevated in the leprosy patients. *IL4R* and *IL21R* both encode immunomodulatory cytokine that regulate adaptive immunity responses, which play important role in the leprosy development. In addition, IL-4 and IL-21 could effect on the proliferation and differentiation and subsequently activate B cells[24].

HLA allele has long been thought to play key role on the development of leprosy. Early studies comparing allele frequencies in leprosy patients and controls have identified associations in both HLA class I gene and class II gene, but results were

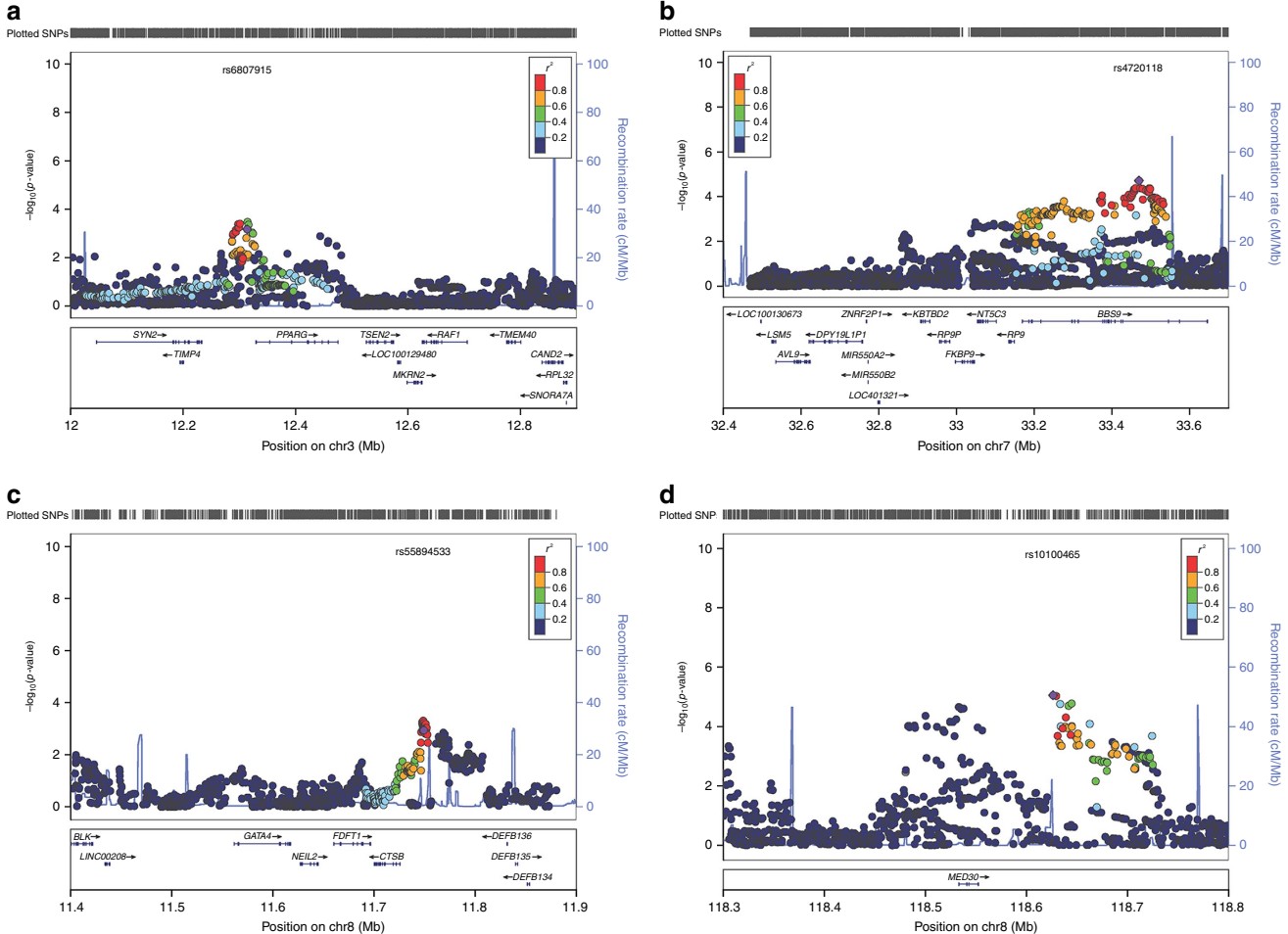

**Figure 2 | Recombination plots of the novel loci reaching genome-wide significance.** (**a**) Association at 3p25.2 (between *SYN2* and *PPARG*); (**b**) Association at 7p14.3 (within *BBS9*); (**c**) Association at 8p23.1 (Nearby *CTSB*); (**d**) Association at 8q24.11 (Nearby *MED30*). SNP labelled in purple is the validated SNP. All coordinates are based on NCBI build 37.

inconsistent. Through HLA imputation, we have refined our previous findings of HLA-DRB1*15 to a four-digit resolution HLA-DRB1*1501, which is consistent with the previously HLA association analysis in Indian[25,26] and Thai[27] populations. Other previously reported associations, such as protective allele HLA-DRB1*04 to Brazilian[28], has been confirmed as suggestive association and similar effect in the current study. HLA-DRB1*1501 has been reported to associate with several autoimmune disease, such as multiple sclerosis[29] and SLE (ref. 30), which emphasize the pleotropic role of disease associated gene between infectious disease and autoimmune disease.

Although GWASs have been proven successfully in identifying regions of the genome harbouring variants that contribute to complex diseases, there are several limitations. First, GWAS have generally identified common risk variants with relatively small effect sizes (OR < 1.5), which are lack of clinical translation but can help to identify important biological pathways for diseases. Second, relying on LD-based association analysis, GWAS is not well powered for detecting rare variants that may have larger effect on disease. Third, for most diseases the effects of all the identified susceptibility loci only account for a small proportion of the estimated heritability, even in diseases where extremely large sample size had been analysed, such as leprosy. By using GCTA method and assuming the leprosy disease prevalence as 0.0001, we estimate the heritability of leprosy attributable to genome-wide SNPs to be 0.199 on the liability scale. This estimate differs from the results from previous genetic epidemiology studies, due to the exclusion of non-additive genetic effects, as well as the effect of gene-environment interaction on leprosy. Finally, the loci identified by GWAS are largely located within non-coding genomic regions, which make it challenging to narrow down causal genes and perform further functional experiments.

Interestingly, all the identified leprosy susceptibility gene/loci at genome-wide significance can only explain ~13.53% of phenotypic variance on the liability scale, which is about 0.199 of the heritability that can be genome-wide SNPs. By conducting the heritability partitioning of leprosy GWAS by tissue and functional category using LD score regression, we identified the most significant enrichment in immune cells and transcript start site, suggesting the important role of immunoregulatory and non-coding variants, but remain further experimental methods to fine-map the causal variant(s).

In summary, we have conducted the largest Genome-wide meta-analysis study on leprosy in the Chinese population to date. By analysing a total number of 8,156 leprosy patients and 15,610 healthy controls, we identified four novel loci and two suggestive loci, which has added new knowledge on the genetic basis of leprosy susceptibility.

## Methods

**Study subjects.** We designed a three-stage case-control analysis for this study. All individuals were of Chinese descent and detailed information for each stage is shown in Supplementary Table 1. All the cases and controls were recruited with the

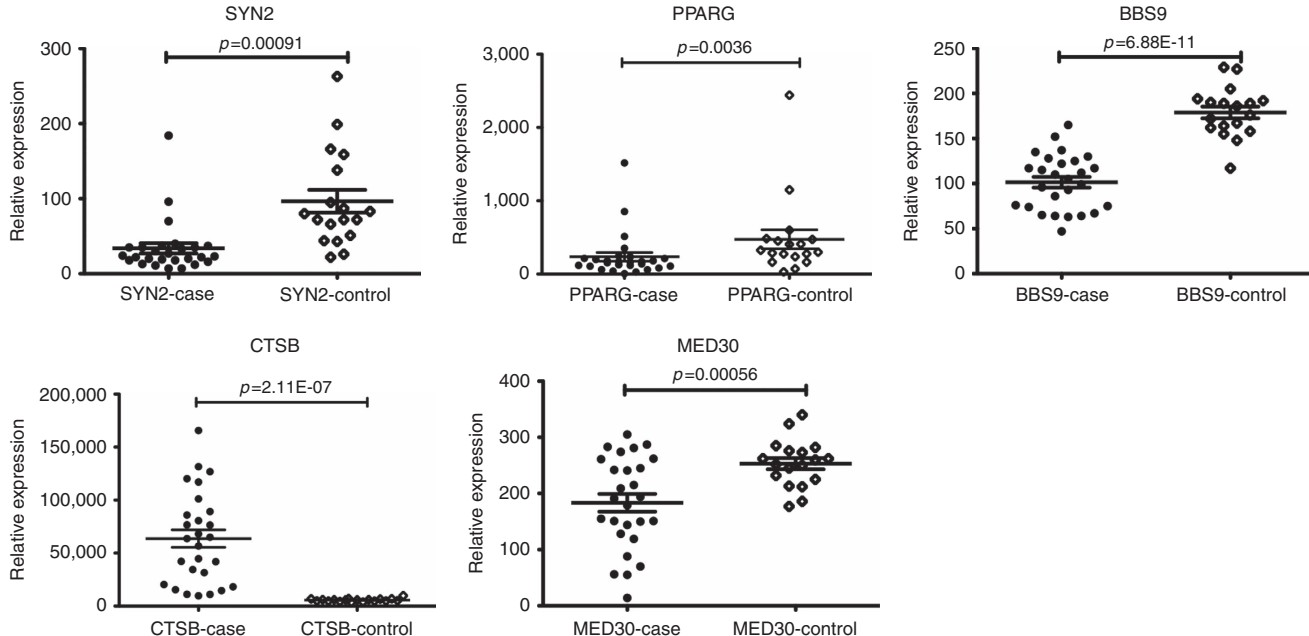

**Figure 3 | Relative gene expression of susceptibility genes nearby the lead association.** The transcripts of these encompassed genes in the susceptibility region measured by RNA sequence data (*SYN2* and *PPARG* on 3p25.2, *BBS9* on 7p14.3, *CTSB* on 8p23.1 and *MED30* on 8q24.11). Statistical significance was calculated by Student's *t*-test comparing leprosy lesion with healthy control. The horizontal line represents significant expression ($P < 0.05$). The dots represent the enroled samples. Data is shown as mean ± s.d.

same criteria described in our previous studies[8–12]. The institutional review board committees of Shandong Provincial Institute of Dermatology and Venereology, Shandong Academy of Medical Science approved our study.

The discovery study (Stage 1) consisted three independent GWAS datasets. The sample characteristics of the first two datasets, GWAS1 (706 leprosy patients and 1,225 healthy controls) and GWAS2 (842 leprosy patients and 925 healthy controls), were described in our previous publications[8–12]. The third was a new GWAS dataset (GWAS3) comprising 1,353 leprosy patients and 1,651 controls of Chinese Han descent, including 901 patients and 899 controls from northern part of China and 452 patients and 752 controls from southern part of China.

Validation analyses were performed in two independent cohorts (Stage 2 and 3): in stage 2, 1,516 leprosy patients and 1,512 controls from northern part of China were analysed; in stage 3, 3,897 leprosy patients and 10,525 controls were analysed, consisting of 1,666 cases and 8,259 controls from Shandong Province of northern China, 829 cases and 589 controls from Yunnan Province, 496 cases and 799 controls from Guizhou Province and 906 cases and 878 controls from Sichuan Province (Yunnan, Guizhou and Sichuan are all from southeastern part of China). All the samples in stage 2 and 3 are Chinese Han descent. Totalling 5,413 cases and 12,037 controls were used in the validation stages.

**Genome-wide genotyping and quality control in the discovery stage.** The genotyping and quality control (QC) procedures of the first (GWAS1) and the second (GWAS2) datasets have been previously described[8,12]. These procedures were conducted on the new (GWAS3) dataset using Illumina Omni Zhonghua chips (Illumina, Inc., San Diego, CA, USA) and standard QC procedures with the following criteria: We excluded all the SNPs with an overall call rate < 95% (1,262 SNPs), minor allele frequency (MAF) < 1% (62,840 SNPs), Hardy-Weinberg equilibrium (HWE) $P$ value in control subjects < $1.0 \times 10^{-8}$ (650 SNPs). We also excluded all the copy number variations (CNVs), intensity-only SNPs and SNPs located in the idiochromosome (a total of 27,754 SNPs) and SNPs with undetermined clusters (three SNPs). Finally, we used 258,961 overlapped genotyped SNPs in the first three independent datasets for imputation and association analysis.

We conducted sample QC in all individuals of GWAS1, 2 and 3. Those with call rate < 96% (8 samples) were excluded first. The potential genetic relatedness of all successfully genotyped samples was estimated on the basis of pairwise identity by state. In total, 241 samples (70 with first-degree familial relationships and 171 with second-degree familial relationships) were removed. The rest samples were tested for population stratification with the principal components stratification method, resulting in the exclusion of 137 population outliers. Finally, a total of 2,743 leprosy patients and 3,573 controls (706 cases and 1,223 controls in GWAS1, 840 cases and 924 controls in GWAS2 and 1,197 cases and 1,426 controls in GWAS3) passed the sample QC filters and were used in subsequent analyses.

**Phasing and imputation.** The SHAPEIT version 2 (ref. 31) was used to conduct phasing analysis on the basis of the common SNPs, and separately for each

ancestry groups. The SNP imputation was carried out with IMPUTE (ref. 32) version 2.2.2 software and 1,000 Genomes Project Phase I reference panel (March 2012 release) in NCBI Build 37 (hg19) coordinates. We subsequently analysed only those SNPs that could be imputed with high confidence (info score $r^2 > 0.8$), had a MAF more than 1% in all samples and without significant deviation from HWE in the controls ($P < 1 \times 10^{-5}$). In total, 5,287,069 imputed SNPs and 258,961 genotyped SNPs passed QC and were tested in the association analysis.

**Statistical analysis.** We first performed the association analysis in three independent GWAS datasets separately. All analyses were carried out using SNPTEST version 2.4.1 software[33]. For each dataset, we included the selected principal components as covariates (1.026 for GWAS1, 1.019 for North Han of GWAS2, 0.98 for South Han of GWAS2, 1.052 for the Southern Minority of GWAS2, 1.038 for North Han of GWAS3 and 1.066 for South Han of GWAS3) in the association model to account for population stratification by EigenCorr[34].

In the discovery stage, We then performed the meta-analysis of the three independent GWAS datasets using the inverse variance method implemented in META (ref. 35) version 1.3.2. The regional association plots for each locus was generated by the online method LocusZoom[36] version 1.3.

In the validation stages, log-additive association testing of SNPs was performed using PLINK v1.07. The meta-analysis of the combined discovery and validation samples of 8,156 cases and 15,610 controls was performed using a fixed effects model (inverse variance method). Cochran's Q statistics were performed to evaluate the significance of heterogeneity among individual studies and Bonferroni-corrected heterogeneity $P$ values of < 0.05 were considered significant.

To check whether additional independent association existed within the identified loci, conditional logistic repression analyses were performed by either SNPTEST (Stage 1) or PLINK (Stage 2 and 3).

**Genotyping and quality control of the selected SNPs in the validation stage.** To validate novel signals identified by the discovery analysis, we applied two strategies to select the potential independent associations. The top SNPs from independent novel loci that showed suggestive association at $P < 5 \times 10^{-4}$ in the newly added discovery sample (GWAS 3), or in the meta-analysis results of combined discovery datasets (GWAS $1 + 2 + 3$) were selected.

Genotyping of the two independent validation stages was conducted using the Sequenom MassARRAY system (Agena Bioscience, Shanghai, China) and TaqMan custom genotyping assays on a 7900 HT Fast Real-Time PCR System (Applied Biosystems, Foster City, CA, USA) according to the manufacturers' instructions. In total, 167 SNPs were selected for stage 2, and 21 were selected for stage 3 using these methods. Of the 21 SNPs in Stage 3, four SNPs failed in genotyping analysis, as they were rejected in the design process (three SNPs) or had a bad genotyping cluster (one SNP). Therefore, they were re-genotyped by using TaqMan custom genotyping assays. In validation analysis, we removed SNPs with a call rate < 90% or undetermined clusters, and samples with call rate < 95%.

**HLA imputation.** We imputed dense SNPs, as well as classical HLA alleles (HLA-A, HLA-B, HLA-C, HLA-DQA1, HLA-DQB1, HLA-DRB1, HLA-DPA1 and HLA-DPB1) and coding variants across the HLA region (chr6: 29.0–33.0 Mb, hg19) in the discovery stage studies using SNP2HLA (ref. 37). Imputation was based on a reference panel from the Pan-Asian[38] array and consisted of genotypes from individuals of Asian descent who were typed for classical HLA 4-digit alleles. The log-additive regression model was used to perform association analysis for all the variants. We used the principal components identified in each discovery data set to correct for population stratification. The final results were generated by fixed effects model meta-analysis.

**Gene annotation and prioritization.** The gene prioritization strategies were based on two methods: (1) differential gene expression analysis, which estimated the relative gene expression difference between leprosy biopsy and healthy control skin through an unpublished RNA-sequence dataset (27 leprosy biopsies versus 18 healthy controls). The genes in closest physical proximity to the lead variant were taken into consideration as potentially causal. (2) eQTL analyses were based on the annotation tool HaploReg v4.1, which accounts for the effect of SNPs on expression from multiple eQTL studies. SNPs with an LD $r^2 > 0.9$ and $D' > 0.9$ were also considered.

**Heritability and enrichment analyses.** For the heritability analysis, we used the method that was implemented in the GCTA software[39] to estimate the contribution of common SNPs. The genetic similarity matrix was estimated using all genotyped autosomal SNPs with a MAF of $> 0.05$ in all the discovery datasets. We used the default option (restricted maximum likelihood, REML) to fit the appropriate variance components model. We assumed that the leprosy prevalence as 0.0001 to estimate the heritability on the liability scale. We also conducted the heritability partitioning of leprosy GWAS by tissue and functional category using the LD score regression.

**Data availability.** The genome-wide association SNP results are available upon request by contacting F.Z at zhangfuren@hotmail.com. Any additional data (beyond those included in the main text and Supplementary Information) that support the findings of this study are also available from the corresponding author upon request.

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

## Acknowledgements

We thank the individuals who participated in this project and Shandong Computer Science Center (National Supercomputer Center in Jinan), which provided us with the platform for statistical analysis. This work was funded by grants from the National Natural Science Foundation of China (81472869, 81402593, 81573036, 81502736, 81620108025), the National Clinical Key Project of Dermatology and Venereology, the Shandong Provincial Independent Innovation Project (ZR2015HZ001), Shandong Province independent innovation and achievement transformation project (2014CGZH1307), the Shandong Provincial Advanced Taishan Scholar Construction Project, the Innovation Project of Shandong Academy of Medical Science, the Natural Science Foundation of Shandong Province (ZR2013HQ041, ZR2014YL044, BS2015YY042, 2014ZRC03145, ZR2015PH027, ZR2015YL035, ZR2015PH040), the Shandong Provincial Medical and Health Development Project (2014GSF118001, 2014WS0064).

## Author contributions

F.Z. conceived of this study and obtained the financial support. F.Z. and H.L. designed the study. H.L., Q.Y., X.F., F.B. undertook recruitment and collected phenotype data.

H.L., Z.W., G.Y., X.F., C.W., Y.X., F.B., Z.Y., J.L., L.S., Y.Y., M.C., Z.M., H.W., P.H., Q.Y., N.W., J.C. conducted sample selection and performed the genotyping of all samples. T.D., W.Y., Y.X., H.X., J.Y., J.L., P.S., G.N., X.C., X.W., X.C. contributed to DNA extraction and clinical data collection. Z.W., Y.S, Y.L., A.I., J.L. undertook data checking, statistical analysis and bioinformatics analyses. H.L. was responsible for sample selection, genotyping and project management. Y.S. and Z.W. wrote the first draft. All authors contributed to the final manuscript, with F.Z., L.H., Z.W., Y.S., X.F. and G.Y. playing the key roles.

## Additional information

**Competing financial interests:** The authors declare no competing financial interests.

**Publisher's note**: 

