## [Peer Review File · Nature Communications]

Reviewers' comments:

Reviewer #1 (Remarks to the Author):

The manuscript describes a genome-wide association study on leprosy using several large, independent sample sets from China that resulted in four novel SNP markers associated with the disease. Within the realm of GWAS - with all the intrinsic limitations that this particular design for association studies holds - the study is very robust; in fact, to my knowledge, this is the largest, most comprehensive association study ever performed for leprosy. Methodological decisions regarding genotype generation and quality control are impeccable. Strategy for data analysis is also very sophisticated, well designed and conducted, leaving virtually no room for discussion on design/methodological decisions and/or procedures. This reviewer congratulates the authors for such achievement.

Yet, upon careful reading of the manuscript, a few possible opportunities for improvement were identified, mainly concentrated in the "Discussion" and related to limitations inherent to the GWAS design, as listed below; for these, any comments from the authors would be appreciated:

1. The introduction brings a statement that the "understanding of the genetic architecture of leprosy has increased rapidly due to the previous GWAS...": it seems that the authors are neglecting a vast body of high quality experimental evidence of involvement of genes in the pathogenesis of leprosy produced in the pre-GWAS era (such as PARK2, IL10, VDR, among others). The quality of the introduction would increase upon the inclusion of some of these seminal findings;
2. The authors state that the findings of the present study "significantly expand" the understanding of genetics of leprosy; however, according to their own data, as stated in the discussion, the heritability of leprosy in the Chinese has increased "... approximately 1,5%...", adding up to a total of 18.5%. It seems excessively optimistic to consider an increase of 1.5% as a significant expansion; please consider rephrasing;
3. Description of the population samples is somewhat superficial; analysis of the data presented in supplementary table 1 reveals both positive (controls older than cases, for example) and intriguing characteristics, such as the gender bias across both cases and controls. The male/female ratio among cases is 3.24 as compared to 1.64 among the controls. Since gender is a classic risk factor for leprosy, did the authors considered adjusting for this variable in the analysis?
4. The "Results" session starts repetitive, with the first paragraph and about half of the second being actually best fitted in "Methods"; please consider reviewing;
5. What was the rationale for choosing (as one out of two strategies) the top SNPs from novel loci showing association only in the newly added GWAS3 sample? What was the role of the study performed using GWSA1 and GWSA2 in the selection of SNPs for the validation phase of the study?
6. Interestingly, all newly identified association signals came from non-coding, intronic or inter-genic SNPs. Though the authors have the merit of applying a clever strategy of combining expression analysis to define the most likely candidate genes, this finding should be better explored in the (somewhat bureaucratic) discussion: what are the limitation of GWAS, even when performed impeccably using extremely large samples? What is the possible impact of decisions such as the inclusion of only common variants for genotyping? What does it mean that the associations identified have very limited impact over risk of disease, as estimated by the OR? As mentioned above, what does it mean that such a powerful study adds only 1.5% on the heritability in leprosy, given that classic, descriptive epidemiology studies (such as twin studies and complex segregation analysis) indicate a much stronger genetic component controlling susceptibility to leprosy? Could the limitation be in part on the choice of a too complex phenotype (disease per se)?
7. As a minor detail: *M. leprae*/*Mycobacterium leprae* should be italicized.

Reviewer #2 (Remarks to the Author):

Authors conducted a genome-wide meta-analysis study of leprosy, one of the chronic infectious diseases. The study included $\sim 1,200$ cases and $\sim 1,400$ controls in the GWAS stage, and $\sim 8,200$ cases and $\sim 15,750$ controls in the replication stage (all were Chinese ancestry). As a result, they reported four novel loci. This is a typical GWAS study, and most of the routinely-used analytic methods are well conducted, including genotype imputation.

1. Since the strongest association signal was observed in the MHC region, risk variant fine mapping using HLA imputation should be applied.
2. As for risk fraction analysis, this reviewer considers that Nagelkerke's pseudo R^2 is different from genetic heritability in terms of both genetics and statistics. GCTA estimation should be applied.
3. Application of LD score regression and functional partitioning would better be applied.

REVIEWERS' COMMENTS:

Reviewer #1 (Remarks to the Author):

The authors consistently addressed all the concerns raised on my first review; it is my opinion that the manuscript is now suitable for publication.

Reviewer #2 (Remarks to the Author):

Authors satisfactorily revised the manuscript, including HLA imputation analysis, heritability estimate, and LD score regression.

Reviewers' comments:

Reviewer #1 (Remarks to the Author):

The manuscript describes a genome-wide association study on leprosy using several large, independent sample sets from China that resulted in four novel SNP markers associated with the disease. Within the realm of GWAS - with all the intrinsic limitations that this particular design for association studies holds - the study is very robust; in fact, to my knowledge, this is the largest, most comprehensive association study ever performed for leprosy. Methodological decisions regarding genotype generation and quality control are impeccable. Strategy for data analysis is also very sophisticated, well designed and conducted, leaving virtually no room for discussion on design/methodological decisions and/or procedures. This reviewer congratulates the authors for such achievement.

Response: Thank you very much for your favorable comments.

Yet, upon careful reading of the manuscript, a few possible opportunities for improvement were identified, mainly concentrated in the "Discussion" and related to limitations inherent to the GWAS design, as listed below; for these, any comments from the authors would be appreciated:

1. The introduction brings a statement that the "understanding of the genetic architecture of leprosy has increased rapidly due to the previous GWAS...": it seems that the authors are neglecting a vast body of high quality experimental evidence of involvement of genes in the pathogenesis of leprosy produced in the pre-GWAS era (such as PARK2, IL10, VDR, among others). The quality of the introduction would increase upon the inclusion of some of these seminal findings;

Response: We totally agree with your comments. By improving the quality of the introduction and addressing your comments, we summarized the genetic findings in the pre-GWAS era and revised the introduction accordingly.

2. The authors state that the findings of the present study "significantly expand" the understanding of genetics of leprosy; however, according to their own data, as stated in the discussion, the heritability of leprosy in the Chinese has increased "... approximately 1,5%...", adding up to a total of 18.5%. It seems excessively optimistic to consider an increase of 1.5% as a significant expansion; please consider rephrasing;

Response: Thanks for the comments. We agree with your comments and revised the phrase accordingly.

3. Description of the population samples is somewhat superficial; analysis of the data presented in supplementary table 1 reveals both positive (controls older than cases, for example) and intriguing characteristics, such as the gender bias across both cases and controls. The male/female ratio among cases is 3.24 as compared to 1.64 among the controls. Since gender is a classic risk factor for leprosy, did the authors considered adjusting for this variable in the analysis?

Response: Thank you very much for your comments. We have done the analysis by adjusting the gender and age for these four significant SNPs. The ORs are almost identical after adjusting for age and gender, suggesting minimal confounding of these factors in the association analysis.

SNP info			Before adjust	After adjust
			gender and age	gender and age
SNP	Chr/Bp	Gene	OR	OR
rs10100465	chr8:118626279	MED30	0.85	0.84
rs4720118	chr7:33469241	BBS9	1.16	1.16
rs55894533	chr8:11749242	CTSB	1.15	1.15
rs6807915	chr3:12313846	SYN2-PPARG	0.89	0.89

4. The "Results" session starts repetitive, with the first paragraph and about half of the second being actually best fitted in "Methods"; please consider reviewing;

Response: As suggested, we have revised part of the results session by editing the repetitive section between Methods and Results.

5. What was the rationale for choosing (as one out of two strategies) the top SNPs from novel loci showing association only in the newly added GWAS3 sample? What was the role of the study performed using GWSA1 and GWSA2 in the selection of SNPs for the validation phase of the study?

Response: 1) As illumina Zhonghua Array was designed for Chinese population specifically, it was consist of thousands of population-specific SNPs. Thus, we selected candidate SNPs by applying GWAS3 as separate dataset. 2) By combining GWAS1+GWAS2+GWAS3, we got a larger sample size that allow us to pick up those SNPs do not reach the selection criteria at previous sample size.

6. Interestingly, all newly identified association signals came from non-coding, intronic or

inter-genic SNPs. Though the authors have the merit of applying a clever strategy of combining expression analysis to define the most likely candidate genes, this finding should be better explored in the (somewhat bureaucratic) discussion: what are the limitations of GWAS, even when performed impeccably using extremely large samples? What is the possible impact of decisions such as the inclusion of only common variants for genotyping? What does it mean that the associations identified have very limited impact over risk of disease, as estimated by the OR? As mentioned above, what does it mean that such a powerful study adds only 1.5% to the heritability in leprosy, given that classic, descriptive epidemiology studies (such as twin studies and complex segregation analysis) indicate a much stronger genetic component controlling susceptibility to leprosy? Could the limitation be in part on the choice of a too complex phenotype (disease per se)?

Response:

We agree with your comments and revised the discussion accordingly. The following paragraph was added to the discussion.

“Although GWASs have been proven successfully in identifying regions of the genome harboring variants that contribute to complex diseases, there are several limitations. Firstly, GWAS have generally identified common risk variants with relatively small effect sizes ($OR < 1.5$), which lack of clinical translation but can help to identify important biological pathways for diseases. Secondly, relying on LD-based association analysis, GWAS is not well powered for detecting rare variants that may have larger effect on disease. Thirdly, for most diseases the effects of all the identified susceptibility loci only account for a small proportion of the estimated heritability, even in diseases where extremely large sample size had been analyzed, such as leprosy. Using GCTA and assuming a disease prevalence of 0.0001, we estimate the heritability of leprosy (the proportion of phenotypic variance attributable to genome-wide SNPs) to be 13.5% on the liability scale. This estimate differs from the results

from previous genetic epidemiology studies, due to the exclusion of non-additive genetic effects as well as the effect of gene-environment interaction on leprosy. Finally, the loci identified by GWAS are largely located within non-coding regions of genome, which makes it challenging to narrow down causal events and perform further functional experiments.

Interestingly, all the confirmed genetic susceptibility loci that have been discovered at genome-wide significance can explain approximately about 13.53% of phenotypic variance on the liability scale, which is about 19.9% of the heritability that can be genome-wide SNPs. By conducting the heritability partitioning of leprosy GWAS by tissue and functional category using LD score regression, we identified the most significant enrichment in immune cells and transcript start site, suggesting the important role of immunoregulatory and non-coding variants, but remain further experimental methods to fine-map the causal variant(s).”

Response to the last question:

A too complex phenotype will definitely increase the genetic heterogeneity and decrease the power of detection of association. One example is major depressive disorder (MDD), for which the selection of a phenotype presents a dilemma for the researchers. The largest genetic study of MDD, including 18,759 subjects in the discovery phase and 57,478 in the replication phase, failed to find any locus reached genome-wide significance association (*Major Depressive Disorder Working Group of the Psychiatric GWAS Consortium. A mega-analysis of genome-wide association studies for major depressive disorder. Molecular psychiatry. 2013;18(4):10.1038/mp.2012.21*). For leprosy, the selection of phenotype is clearly according to the diagnosis criteria, thus several susceptibility gene were successfully identified.

7. As a minor detail: *M. leprae*/*Mycobacterium leprae* should be italicized.

Response: Thanks very much for pointing it. We have revised it accordingly in the revision.

Reviewer #2 (Remarks to the Author):

Authors conducted a genome-wide meta-analysis study of leprosy, one of the chronic infectious diseases. The study included ~1,200 cases and ~1,400 controls in the GWAS stage, and ~8,200 cases and ~15,750 controls in the replication stage (all were Chinese ancestry). As a result, they reported four novel loci. This is a typical GWAS study, and most of the routinely-used analytic methods are well conducted, including genotype imputation.

1. Since the strongest association signal was observed in the MHC region, risk variant fine mapping using HLA imputation should be applied.

Response: Thanks very much for the great suggestion. We have finished the HLA imputation analysis in the discovery stage studies by using SNP2HLA and based on a reference panel from the Pan-Asian population. We found that the extensive associations within the MHC region were driven by variants located around the region containing the HLA class II genes, with HLA-DRB1*15:01 identified as the most significant HLA variant ($P = 4.21 \times 10^{-44}$; OR = 2.17). Conditioning on HLA-DRB1*15:01 could eliminate the strong and extensive associations within the MHC region. The full association results of classical HLA allele were provided in the **[supplementary table 5]**. We also added relevant results, discussion as well as methods part in the revised manuscript.

2. As for risk fraction analysis, this reviewer considers that Nagelkerke's pseudo R2 is

different from genetic heritability in terms of both genetics and statistics. GCTA estimation should be applied.

Response: Thanks very much for the suggestion. We estimated the proportion of variance in liability to leprosy explained by SNPs using GCTA (Genome-wide Complex Trait Analysis), which uses genome-wide SNP genotypes to calculate heritability in the population. We estimate the SNP heritability of leprosy at 19.9% (se = 0.01), by using the genotyped autosomal SNPs and assuming a disease prevalence of 0.0001 on the liability scale. In total, all the identified Genome-wide significant variants thus far as being robustly associated with leprosy risk explain approximately 13.53% (se = 0.01) on the liability scale.

Supplementary Table 6 Heritability estimates for genome-wide SNPs in leprosy on different disease risk.

Category	All SNPs		known region (LD) explained ratio
	Heritability	SE	
Liability scale h ² (prevalence = 0.0001)	0.199	0.010	13.53%

3. Application of LD score regression and functional partitioning would better be applied.

Response: We have conducted the heritability partitioning of leprosy GWAS by tissue and functional category using LD score regression identified significant enrichment in multiple tissue. The most significant enrichment was found in immune cells (enrichment = 3.74, $p = 3.2 \times 10^{-8}$), suggesting the important role of immunity in the disease etiology. Regions of functional category of genome were most significantly enriched in transcription start site (TSS, enrichment = 16.3, $p = 2.52 \times 10^{-4}$) [Supplementary Figure 4].

Supplementary Figure 4 Overall genetic architecture of leprosy across functional categories and tissues.

Enrichment estimates for the main annotations and tissues of LDSC. Error bars represent 95% confidence intervals around the estimate. Categories are sorted by P value, with boxes indicating annotations or tissues that pass the multiple testing significance thresholds. CNS, central nervous system; DHS, DNase hypersensitivity; GI, gastrointestinal; TFBS, transcription factor binding site; Tss, transcription start site; UTR, untranslated region.

REVIEWERS' COMMENTS:

Reviewer #1 (Remarks to the Author):

The authors consistently addressed all the concerns raised on my first review; it is my opinion that the manuscript is now suitable for publication.

Response: We thank the reviewer for their helpful comments to improve our manuscript.

Reviewer #2 (Remarks to the Author):

Authors satisfactorily revised the manuscript, including HLA imputation analysis, heritability estimate, and LD score regression.

Response: We thank the reviewer for their helpful comments to improve our manuscript.